# Slab tearing and its surface signals controlled by passive margin strength

Giridas Maiti [1], Nevena Andrić-Tomašević [1], Attila Balázs [2], Lucas H. J. Eskens[1] & Taras Gerya[2]

Slab tearing, the lateral detachment of subducting oceanic slab from continental lithosphere, is widely inferred from seismic tomography, yet its surface expressions in mountain belts and adjacent foreland basins remain ambiguous and often contradictory. Existing geodynamic models predict that slab tearing propagates at unrealistically high velocities, implying its transient signatures unlikely to be preserved in surface or stratigraphic records. In contrast, geological observations, such as lateral migration of foreland basin depocenters and systematic basin thickening in the direction of tear propagation, indicate more persistent surface responses, highlighting a long-standing disconnect between models and field evidence. Here we resolve this paradox by showing that lateral variations in passive-margin strength fundamentally control the initiation, propagation, and surface imprint of slab tearing. Using fully coupled three-dimensional thermo-mechanical and surface-process simulations, we demonstrate that accounting for passive-margin heterogeneity significantly slows tear propagation and produces long-lived tectonostratigraphic signatures consistent with natural examples from the Alps, Carpathians, Zagros, and other orogenic belts. These results bridge deep-mantle dynamics and surface geological records, providing a unified framework to identify and interpret slab tearing in orogenic systems worldwide.

Mountain building at continental collision zones is profoundly influenced by subducting slab dynamics. Shortly after the onset of collision, the negatively buoyant oceanic slab typically detaches from the buoyant continental lithosphere in a process known as slab detachment or breakoff[1,2]. Seismic tomography reveals that this detachment usually migrates laterally along strike of collisional boundaries, a process termed in literature as slab tearing[3,4]. Slab tearing is therefore a fundamental geodynamic process in the early stages of orogeny and exerts first-order control on the evolution of mountain belts. Since its recognition, it has been invoked in nearly every major collisional orogen (Fig. 1a), including the Alps–Carpathians[5,6], Apennines[7–9], western Mediterranean[4,10], Aegean–Anatolia–Zagros[11], Caucasus[12], and Himalaya[13,14]. Although seismic tomography data (Fig. 1b, c) supports these interpretations[3,11,12,15], the surface expressions of slab tearing in

orogens[5,13,14] and adjacent foreland basins[6,8,11] remain widely debated and largely speculated[16].

Existing geodynamic models provide important insights into the initiation and dynamics of slab tearing, but they consistently predict a rapid lateral tear-propagation velocity of 35–120 cm yr$^{-1}$[17–23]. These values are an order of magnitude faster than indirect estimates from geological proxies, which are typically <10 cm yr$^{-1}$[6–8]. Such rapid propagation in earlier models yields only short-lived surface signals, which are difficult to identify and separate from collisional overprints. Moreover, previous models largely ignore feedback between tectonics, erosion, and sedimentation that shape coupled mountain–basin systems. This leaves a critical gap in linking deep-lithospheric process slab tearing to preserved surface records. Closing this gap is essential for reliably identifying slab tearing in surface

[1]Institute of Applied Geosciences, Karlsruhe Institute of Technology, Karlsruhe, Germany. [2]Institute of Geophysics, ETH, Zürich, Switzerland. e-mail: giridas.maiti@kit.edu

geological records and for reconstructing the spatial and temporal evolution of mountain building in collisional orogens worldwide.

Foreland basins that form adjacent to growing mountain belts provide a key archive of slab-tearing's surface signals. Their stratigraphy records the interplay between tectonic uplift driven by plate convergence and subduction dynamics, and sediment fluxes modulated by climate, lithology, and sea level[24–27]. Yet isolating slab-tearing signals in these basins is inherently challenging. Because tearing typically occurs just before or during the early stages of collision[3,16,18,19], its imprints are often overprinted by subsequent collisional processes. In addition, features often attributed to slab tearing may also arise from alternative mechanisms, including oblique convergence[28,29], lateral variations in lithospheric flexural rigidity[30], or far-field eustatic effects[31]. Disentangling these competing signals requires models that capture both deep-lithospheric dynamics and surface processes.

A likely reason for unrealistically fast tearing in previous models is the assumption of laterally uniform passive margins. In reality, passive margins are highly heterogeneous, with variations in oceanic age[32], crustal structure, and the presence of microcontinental blocks separated by exhumed mantle or younger oceanic branches[33,34]. Because slab detachment localizes at the ocean–continent transition or along former passive margins[1], these heterogeneities are expected to exert strong control on where tearing initiates, how it propagates, and how its signals are transmitted to the surface. Here, we emphasize that these heterogeneities translate into along-strike variations in passive-margin strength, which we define as integrated lithospheric yield strength governed by crustal thickness, thermal state, composition, and adjacent oceanic age. We therefore hypothesize that variations in passive-margin strength govern tear-propagation velocity and control the generation and preservation of diagnostic tectonostratigraphic signals in foreland basins adjacent to orogenic belts, which our models evaluate against the Northern Alpine Foreland Basin (NAFB) record. To test this hypothesis, we conduct state-of-the-art, fully coupled 3D thermo-mechanical and surface-process simulations that incorporate realistic passive-margin heterogeneity (Supplementary Fig. S1). The models simulate slab tearing beneath the Alps between ~30–20 Ma, when closure of the Piemont–Ligurian Ocean led to collision of the European passive margin with the Adriatic plate, triggering the Alpine orogeny[4,5,35].

Here, we show that along-strike variability in passive-margin structure, marked by the Briançonnais terrane[36,37] and contrasting oceanic ages of the Valais and Piemont–Ligurian domains[38,39], provides natural conditions for passive-margin strength heterogeneity. This heterogeneity controls the location of tear initiation and produces tear-propagation velocities comparable to geological estimates during Alpine collision. Our results reveal that slab tearing generates coeval but contrasting depositional environments in foreland basins, drives lateral migration of these environments with the advancing tear, and produces progressive basin thickening in the direction of tear propagation. By isolating the transient signals of slab tearing, evaluating their preservation potential, and comparing them with the stratigraphic record of the NAFB[6,40,41] and other foreland basins worldwide, we provide diagnostic signatures for identifying slab tearing in surface geological records and refining models of orogenic evolution.

## Results

Slab tear initiation in the models begins locally above the microcontinental block. Entry of this buoyant, mechanically weaker continental lithospheric block into the subduction zone at ~20 Myr causes slab necking, leading to detachment at ~22 Myr (Fig. 2a–c). Importantly, detachment does not occur simultaneously along strike of the collisional margin. Instead, it starts in the microcontinent region, where rheological strength is lowest (Supplementary Fig. S2), and then propagates laterally, with passive-margin strength variations controlling both the rate of propagation and the migration of surface responses. This leaves the earliest uplift/subsidence signals of slab tear spatially localized.

The slab detachment at the microcontinent induces isostatic rebound and strong vertical motions in the overlying orogen and foreland (Fig. 3b.i, ii). Orogenic uplift (~2–3 mm yr⁻¹) enhances erosion (~0.4 mm yr⁻¹), supplying sediments to the adjacent foreland (~0.1–0.3 mm yr⁻¹) along orogen-perpendicular transport routes (Fig. 3a.i, ii, 3c.i, ii). Meanwhile, further along strike, the foreland region still attached to the subducting slab continues to subside at 3–4 mm yr⁻¹ (Fig. 2j), driven by slab pull (Supplementary Fig. S3), producing greater accommodation space. This along-strike contrast in uplift and subsidence creates two coeval but contrasting depositional environments in the foreland basin separated by a depositional slope (Fig. 3b.i, ii; Supplementary Fig. 4). Sediment not accommodated in the uplifted foreland bypasses along this slope toward the subsiding, slab-attached domain (Fig. 3a.ii–iii), where accumulation rates reach 0.5–2 mm yr⁻¹ (Fig. 3c.i–ii) despite low adjacent orogenic topography.

By ~26 Myr, detachment in the microcontinent region is followed by exhumation of subducted crust within the wedge (Fig. 2e–f; Supplementary Fig. S5), further elevating orogenic topography. From this stage onward, the subsidence in the slab-detached region of the foreland is increasingly controlled by orogenic load. Elevated topography also enhances erosion rates of ~4.5 mm yr⁻¹, producing high sediment flux to the foreland (Fig. 3c.iii).

As the tear front migrates laterally away from the microcontinent (Fig. 2g–i), it is accompanied by foreland and/or orogen uplift (Fig. 2b–f). This leads to progressive shallowing of depositional environments in the foreland. By ~27.6 Myr, the tear reaches the rear end of the model domain, triggering dynamic uplift in the rear region (Fig. 2k, l). The uplift shallows the rear foreland, leading to reduction in accommodation space (Fig. 2f). Meanwhile, subsidence increases in the frontal foreland due to higher hinterland thrust-induced topographic load (Fig. 2l). Together, these opposing vertical motions tilt the foreland basin axis toward the front of the model domain, leading to an "inversion" of the basin's axial-sediment transport direction (Fig. 3a.iv). During tear propagation, accumulated sediment thickness in the foreland increases in the direction of propagation (Fig. 3d) because regions ahead of the advancing tear remain subsiding longer and continue to receive sediment until the tear arrives.

Over the 20–27.6 Myr interval, the tear propagates ~800 km laterally, with an average velocity of ~10.5 cm yr⁻¹—an order of magnitude slower than previous model predictions[17,19–23] and comparable to geological estimates[6–8]. However, its path is not purely lateral. After reaching the end of the microcontinent, a lateral transition from continental to oceanic lithosphere causes the tear to propagate vertically (Fig. 2i). This vertical-tear phase leaves no significant surface imprint in the foreland basin record, but it slows surface projection of the tear-propagation velocity and prolongs rear-foreland subsidence.

Additional experiments show that tear velocity increases with decreasing oceanic lithosphere age and with reduced along-strike variations in passive-margin strength (Supplementary Table S1 and Figs. S6–S10). Variations in mantle viscosity further modulate tear propagation in our models. A weaker, less viscous mantle promotes rapid tearing, whereas a stronger, more viscous mantle slows slab sinking and reduces tear velocity (Supplementary Table S3 and Figs. S11–S14), consistent with recent findings[42]. Sensitivity tests on microcontinent size and thermal structure in the reference model further show that increasing either microcontinent length, width, or Moho temperature does not substantially affect the onset of slab necking, which begins at ~20 Myr in all models (Supplementary Figs. S17–S22). However, a wider microcontinent delays tearing by prolonging the phase of vertical-tear development at its rear edge, where the tear transitions from the microcontinent to the adjacent oceanic domain (Supplementary Figs. S19 and S20). Interestingly, a warmer microcontinent promotes intraplate deformation and vertical

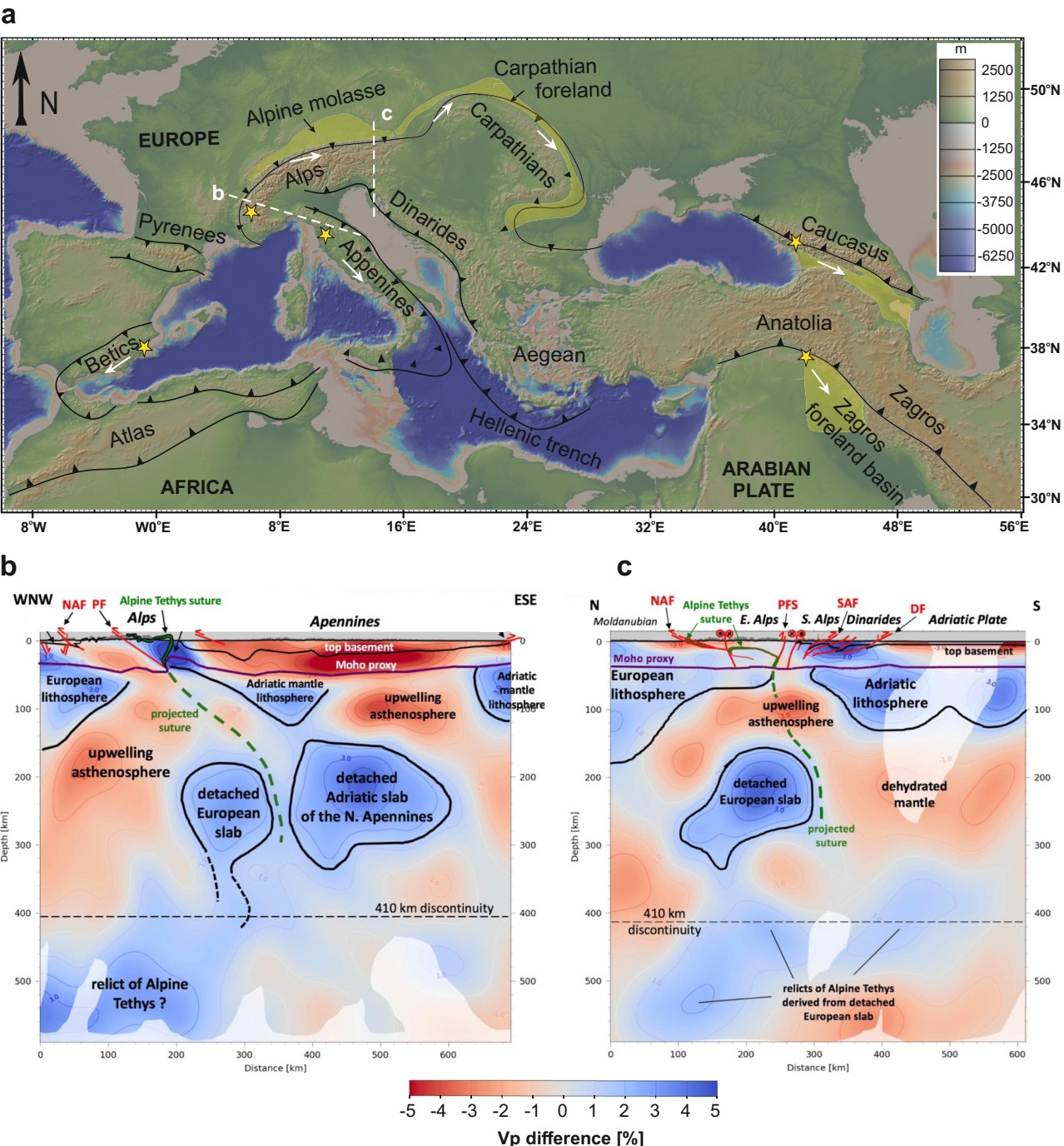

**Fig. 1 | Orogenic belts affected by slab tearing, and an example of a present-day detached slab beneath the Alps. a** Reported slab-tear-propagation direction in orogenic belts. The yellow star marks the initial site of slab detachment, and white arrows indicate the direction of detachment/tear propagation. Yellow shading highlights the affected foreland basins (figure made with GeoMapApp[70] (www.geomapapp.org)/CC-BY). **b, c** Present-day seismic structure beneath the western (profile-b marked in Panel-a) and eastern (profile-c marked in Panel-a) Alps reveals detachment of the subducted slab from the European continental lithosphere, and some relics of the Alpine Tethyan oceanic slab have reached into the slab graveyard within the mantle transition zone (adapted from ref. 15 under CC-BY 4.0). The abbreviations are NAF- Northern Alpine Front, PF- Penninic Front, PFS- Periadriatic Fault System, SAF- Southern Alpine Front, DF- Dinaric Front.

thickening, which prolongs the duration of slab necking and breakoff at the frontal end and thereby slows the overall along-strike tear propagation (Supplementary Figs. S21 and S22). Despite differences in tear-propagation velocity, all models show a similar evolutionary pattern in which basin uplift and shallowing migrate along strike with tear propagation (Supplementary Figs. S6–S14; S17–S22). However, slow tearing prolongs along-strike contrasts in uplift and subsidence and creates strong asymmetric foreland basins (Supplementary Figs. S8,

S9, S13, S14, S19, S20, and S23), whereas fast tearing produces more uniform basin architecture (Supplementary Figs. S6, S7, S10, and S23).

## Discussion

### Coeval but contrasting depositional environments in the foreland basin

Model results indicate that slab tearing induces along-strike variations in slab pull, generating differential uplift and subsidence,

producing coeval but contrasting depositional environments in foreland basins. In the Oligocene–Miocene Northern Alpine Foreland Basin (NAFB, Fig. 4a), this mechanism explains the coexistence of terrestrial to shallow-marine settings in the west, above a detached slab, and deep-marine environment (~1000–1500 m water depth)[41,43] in the east, above the still-attached slab[41,44]. Variable slab pull also created an eastward depositional slope, establishing an axial-sediment-routing system from uplifted western Alps to subsided eastern Alps[44] (Fig. 4a.ii, v). Alternatively, axial drainage may arise from differential flexural subsidence of the foreland due to asynchronous thrust loading during oblique continental collision[28]. However, existing numerical models[45] suggest that such loading typically drives drainage in the opposite direction, toward the foreland adjacent to higher hinterland topography, where flexural subsidence is greatest. The axial-drainage system in the Oligocene–Miocene NAFB, however, developed west-to-east, where the eastern Alps had lower hinterland topography[46]. Moreover, syn-flexural fault growth in the absence of high hinterland topography indicates that subsidence in the eastern NAFB was controlled by slab loading[47]. These observations support our interpretation of Oligocene–Miocene eastward tear propagation and an attached European slab beneath the eastern Alps.

## Increasing depositional thickness in the direction of tear propagation

Model results reveal that as the slab-tear propagates laterally, uplift of foreland and hinterland migrates coevally. Migration of foreland uplift produces time-transgressive shallowing of depositional environments, while hinterland uplift enhances erosion and increases sediment influxes to the foreland (Fig. 3a). The models further indicate that the slab-attached foreland ahead of the advancing tear front remains in subsidence longer, accumulating thicker sedimentary successions (Fig. 3d). Observations from multiple orogenic systems, including the Alps[41], Carpathians[6,48,49], Caucausus[12], Apennines[8], Zagros[11,50], mirror these predictions (Fig. 4b.v). In the Carpathians, late Oligocene-Pliocene[6,51] depocenter migration and thickening toward the east (Fig. 4b.v) indicates west-to-east tear propagation, with depocenter thickness and sedimentation rates[6,48,49] closely matching model predictions (Fig. 4b.iii–v). Similarly, late Miocene uplift–subsidence contrasts in the NW Zagros foreland, attributed to slab tearing, resemble modeled along-strike variations in depositional thickness (Fig. 4b.v). These parallels suggest that slab tearing, as a lithospheric-scale process, leaves broadly consistent stratigraphic fingerprints, including lateral facies contrasts, paleo-flow direction, and increasing depositional-thickness gradients toward tear-propagation direction. However, local mountain erosion rates, source rock erodibility, and collisional architecture may modulate variation in depocenter thickness, sedimentation rate, and timing of imprinting slab-tear signals in foreland records[27,52,53].

## Reversal in the axial-drainage system

Models predict that once slab-tear propagation is completed, then isostatic rebound reverses previously established axial drainage (Fig. 4a.iii, vi). This reversal is amplified due to enhanced flexural subsidence induced by thrust loading in the frontal foreland region, where slab detachment occurred first. There, continued upper-plate convergence and associated crustal exhumation, along with thrusting following slab detachment, increased the topographic loading and thus higher flexural subsidence (Fig. 4a.iii, vi). Such reversal is recorded in the Miocene NAFB, where provenance and stratigraphic shifts indicate drainage reversal at ~18–16 Ma[44]. In contrast, the Carpathian foreland shows no such reversal. There, sediment thickness continues to increase eastward[6], reflecting an attached slab, as also observed in the present-day Vrancea region of the southeastern Carpathians[54].

## Surface expression and preservation potential of slab-tear signals

Whether slab-tearing signatures are fully expressed depends strongly on tear-propagation velocity (Supplementary Figs. S6–S14). Faster tearing shortens the duration of differential uplift–subsidence and limits slab-tear signal expression and produces more uniform along-strike basin architecture (Supplementary Fig. S23). Whereas slower tearing prolongs along-strike contrasts in uplift and subsidence, yields thicker successions toward the tear-propagation direction, and creates strong asymmetric foreland basins (Fig. 4b.v). Tear velocity depends on variations in passive-margin strength, passive-margin architecture, variations in adjacent oceanic lithosphere age, and mantle viscosity (Supplementary Tables S1–S2). By incorporating these heterogeneities, our models yield tear-propagation velocities more consistent with geological observations than earlier homogeneous passive-margin modeling approaches[18,21–23].

Since slab detachment and tearing occur during the early stages of continental collision, as also conceived in the original long-standing hypothesis[1,4], preservation of their surface signatures would depend on subsequent collisional overprint. In rapidly converging or long-lived orogens (e.g., Himalayas[55]), thrusting and burial of foreland deposits within the orogenic wedge may obscure, destroy, or overprint these signals[26,56]. Conversely, younger or more slowly converging systems such as the Alps, Zagros, and Caucasus are more likely to retain them. However, preservation is most favorable in slab-pull-dominated orogens[57], where convergence is primarily driven by the subducting slab's pull rather than upper-plate compression. In such settings, limited upper-plate thrusting minimizes overprinting, allowing slab-tearing signatures to be retained[57].

Additional complexity in tear propagation arises when lateral tearing transitions into vertical tearing. In our models, the presence of a microcontinent triggers a vertical tear at its rear edge, where the tear transitions from the microcontinent segment into the adjacent oceanic domain (Fig. 2i). The vertical-tear phase delays the overall along-strike tear propagation and is expressed at the surface by prolonged subsidence of the rear-foreland basin (Supplementary Fig. 23). We also observe that a wider microcontinent prolongs the phase of vertical tearing and therefore slows the along-strike tear propagation, consistent with previous modeling study[19]. Furthermore, other heterogeneities, such as variations in slab strength along the trench, can cause vertical-tear development[20,58]. Seismic tomography supports the occurrence of such structures, for example, beneath the Apennines[9]. However, these features remain largely absent in many numerical models that assume homogeneous slabs and laterally uniform passive margins. In nature, however, subducting slabs are inherently heterogeneous, as indicated by tomographic observations from the Alps[15] and global oceanic-age datasets[32]. In our 3D thermo-mechanical models, incorporating along-strike passive-margin strength variations together with a microcontinent, as inferred for the Alps, produces tear-propagation velocities comparable to those estimated from Alpine geological records. Earlier numerical studies showed that collisional-margin obliquity can reduce tear-propagation velocity, yet the predicted rates still remained higher than geological estimates because these models mostly assumed homogeneous passive margins[19]. Moreover, margin obliquity may vary along strike and may be very little or absent in some collisional systems. Our results, therefore, show that passive-margin strength heterogeneity can exert a first-order control on lateral tear-propagation velocity, even along a straight or non-oblique margin. This finding highlights the need for future geodynamic models to incorporate natural along-strike variations in passive-margin strength, along-strike oceanic-age variation, structural heterogeneity associated with microcontinents, and collisional-margin obliquity. Including these factors will be essential for reproducing realistic slab-tear dynamics and propagation velocities, and for linking slab tearing more reliably to their surface

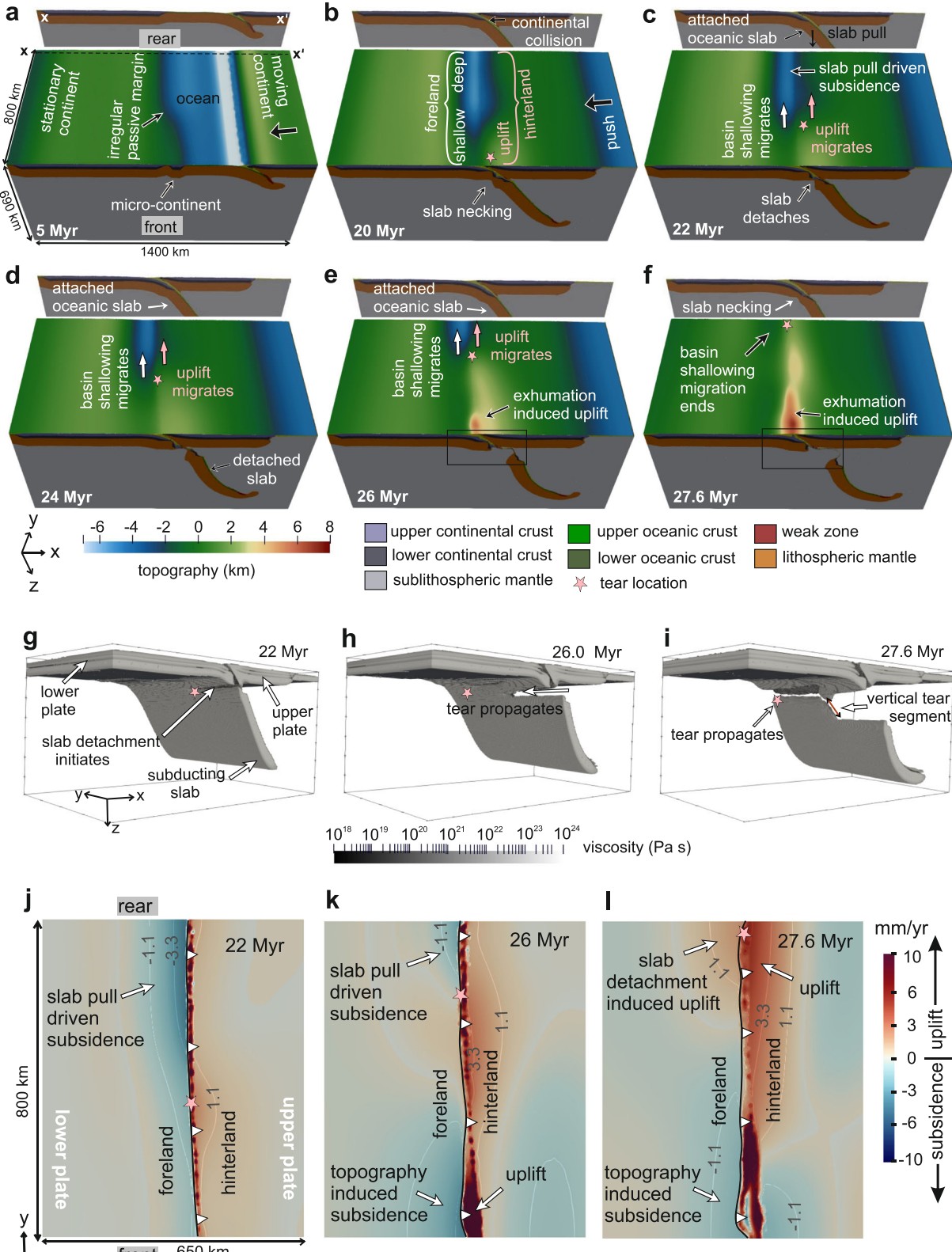

geological records. Altogether, our results provide a plausible explanation for the persistent mismatch between numerically predicted and geologically inferred tear-propagation velocities, and establish passive-margin strength heterogeneity as a key factor in bridging that gap.

Finally, we conclude that slab tearing leaves three diagnostic signatures in the tectonostratigraphic records of foreland basins adjacent to mountain belts: (i) coeval but contrasting depositional environments that migrate laterally with the advancing tear, (ii) progressive basin thickening in the direction of tear propagation, and (iii) reversal of axial drainage (Fig. 5). Together, these features capture the dynamic coupling between slab tearing, mountain uplift, and evolution of adjacent foreland basins. Importantly, the mountain uplift and foreland basin response should be viewed as slab tearing *sensu lato*—affecting regions

**Fig. 2 | Surface topography, slab tearing, and differential subsidence/uplift in the foreland and hinterland during reference-model evolution. a–f** Evolution of model topography (shaded relief), crustal, and lithospheric structure. To show the evolution of the slab at the rear side, we provide a cross-section taken at the rear edge of the model domain (section taken along xx'). Color scale for surface topography and colors representing different compositional layers of the model are shown below. **g–i** Along-strike propagation view of slab tearing. The color bar denotes slab viscosity. We have extracted the slab geometry by plotting the temperature iso-surface $T = 1300\,°C$. **j–i** Spatial patterns of subsidence and uplift rates in the foreland and hinterland. **j, k** show that, at the rear side of the model domain, foreland subsidence is driven by prolonged slab pull (i.e., dynamic subsidence). In contrast, at the front, foreland subsidence is dominated by topographic loading, where early slab detachment builds higher topography. **l** reveals a dynamic uplift in both foreland and hinterland regions when the slab tear reaches the rear end (27.6 Myr). This is expected to give rise to a reversal in sediment flow direction for a short period of time.

above both slab-detached and slab-attached segments—rather than *sensu stricto*, which would restrict interpretation to the narrow zone above the active slab-tear location. By demonstrating how the tearing of subducting slabs shapes mountains and basins over time, our study links mantle-scale processes to their long-lived surface expressions, providing a framework for identifying slab tearing in geological records and refining models of orogenic evolution worldwide.

## Methods

The state-of-the-art numerical modeling approach couples the 3D thermo-mechanical code I3ELVIS[59,60] with a surface-process modeling code FDSPM[61]. The coupling algorithm allows a full two-way feedback between the lithospheric-scale geodynamic (e.g., slab detachment or tearing) and surface processes (e.g., sedimentation and erosion). Below, we provide a detailed explanation of the governing equations and rheological formulation, which can also be found in previous applications of the I3ELVIS code[19,20].

The numerical algorithm of the I3ELVIS code, based on staggered finite-differences and marker-in-cell techniques, solves the mass (1), momentum (2) and energy conservation (3) equations for incompressible media:

$$\frac{\partial v_i}{\partial x_i} = 0 \tag{1}$$

$$\frac{\partial \sigma_{ij}}{\partial x_j} - \frac{\partial P_i}{\partial x_i} = -\rho g_i \tag{2}$$

$$\rho C_{p,\text{eff}} \frac{DT}{Dt} = \frac{\partial}{\partial x_i}\left(k\frac{\partial T}{\partial x_i}\right) + H_r + H_s + H_a \tag{3}$$

$$H_a = T\alpha \frac{DP}{Dt} \tag{4}$$

$$\rho_{T,P} = \rho_0(1 - \alpha(T - T_0))(1 + \beta_c(P - P_0)) \tag{5}$$

where $v$ is velocity, $\sigma$ is the deviatoric stress tensor, $P$ is the total pressure (mean normal stress), $\rho$ is the density, $g$ is the gravitational acceleration, $C_{p,\text{eff}}$ is the effective isobaric heat capacity, $T$ is the temperature, $k$ is the thermal conductivity, which depends on pressure, temperature and rock composition, $H_r$ is radioactive heating that is constant for a given rock composition, $H_s$ is shear heating (product of deviatoric stress and strain rate), $H_a$ is the adiabatic heating (see Eq. 4), $\eta_{\text{eff}}$ is the effective viscosity for non-linear visco-plastic deformation, $\alpha$ is the thermal expansion, and $\beta_c$ is the coefficient of compressibility. The temperature equation is solved in Lagrangian form, and temperature advection is implemented through a marker-in-cell technique. The Einstein notation is used for the indices $i$ and $j$, which denote spatial directions $i = (x, y, z)$ and $j = (x, y, z)$ in 3D. Physical properties are transported by Lagrangian markers that move with the velocity field interpolated from the fixed grid. Detailed thermal and mechanical model parameters are provided in Supplementary Table S1.

### Rheology

Diffusion and dislocation creep flow laws combined with the Drucker–Prager yield criterion[62] were used to determine whether viscous or plastic deformation occurs. These are implemented in the models and account for visco-plastic behavior. The effective creep viscosity $\eta$ of a material is determined as following:

$$\frac{1}{\eta_{\text{eff}}} = \frac{1}{\eta_{\text{diff}}} + \frac{1}{\eta_{\text{disl}}} \tag{6}$$

where $\eta_{\text{diff}}$ and $\eta_{\text{disl}}$ are calculated by using Newtonian diffusion creep and power law dislocation creep, respectively:

$$\eta_{\text{diff}} = \frac{1}{2}A_D^{-1}\sigma_{\text{crit}}^{1-n}\exp\left(\frac{PV_a + E}{RT}\right) \tag{7}$$

$$\eta_{\text{disl}} = \frac{1}{2}A_D^{1/n}\dot{\varepsilon}_{II}^{(1-n)/n}\exp\left(\frac{PV_a + E}{nRT}\right) \tag{8}$$

where $A_D$ is the pre-exponential factor, $n$ is the power law exponent, $V_a$ is the activation volume, $E$ is the activation energy, $R$ is the universal gas constant, $\sigma_{\text{crit}} = 10\,\text{kPa}$ is the transition stress from diffusion to dislocation creep[63] and $\dot{\varepsilon}_{II}$ is the second invariant of strain rate. Plasticity is implemented using the following yield criterion $\sigma_{II} \leq \sigma_{\text{yield}}$, which limits creep viscosity, altogether yielding an effective viscosity limit:

$$\eta_{\text{eff}} \leq \frac{\sigma_{\text{yield}}}{2\dot{\varepsilon}_{II}} = \frac{C_0 + P_e \sin\varphi}{2\dot{\varepsilon}_{II}} \tag{9}$$

where $C_0$ is the cohesion, $\varphi$ is the friction angle, $P_e$ is the effective pressure (total pressure subtracting the hydrostatic fluid pressure), $\dot{\epsilon}_{II}$ is the second invariant of strain rate. Plastic strain weakening is implemented by linearly decreasing the cohesion and friction angle over the strain interval of 0–1. Grain-size–dependent weakening was not included in the model.

### Diffusion-controlled surface processes

Erosion and sedimentation are simulated by coupling a finite-differences surface-process model (FDSPM[61]) to the thermo-mechanical code (I3ELVIS), assuming constant diffusivity and the following advection–diffusion equation:

$$\frac{\partial h}{\partial t} = k\nabla^2 h + v_{\text{vertical}} + \nabla h.\mathbf{v}_{\text{horizontal}} \tag{10}$$

where $h$ is topography, $t$ is time, $\kappa$ is the diffusion coefficient, $v_{\text{vertical}}$ is the vertical velocity component, and $\mathbf{v}_{\text{horizontal}}$ is the surface horizontal velocity vector. On the considered surface, where topographic curvature $\nabla^2 h$ is negative, erosion occurs, and on the surface where $\nabla^2 h$ is positive, sediment deposition takes place. The coupling algorithm between the thermo-mechanical and surface processes codes allows a full two-way feedback. Surface topography in the models is therefore controlled by the balance among diffusive smoothing, vertical tectonic uplift, and horizontal advection. Where uplift and/or advection

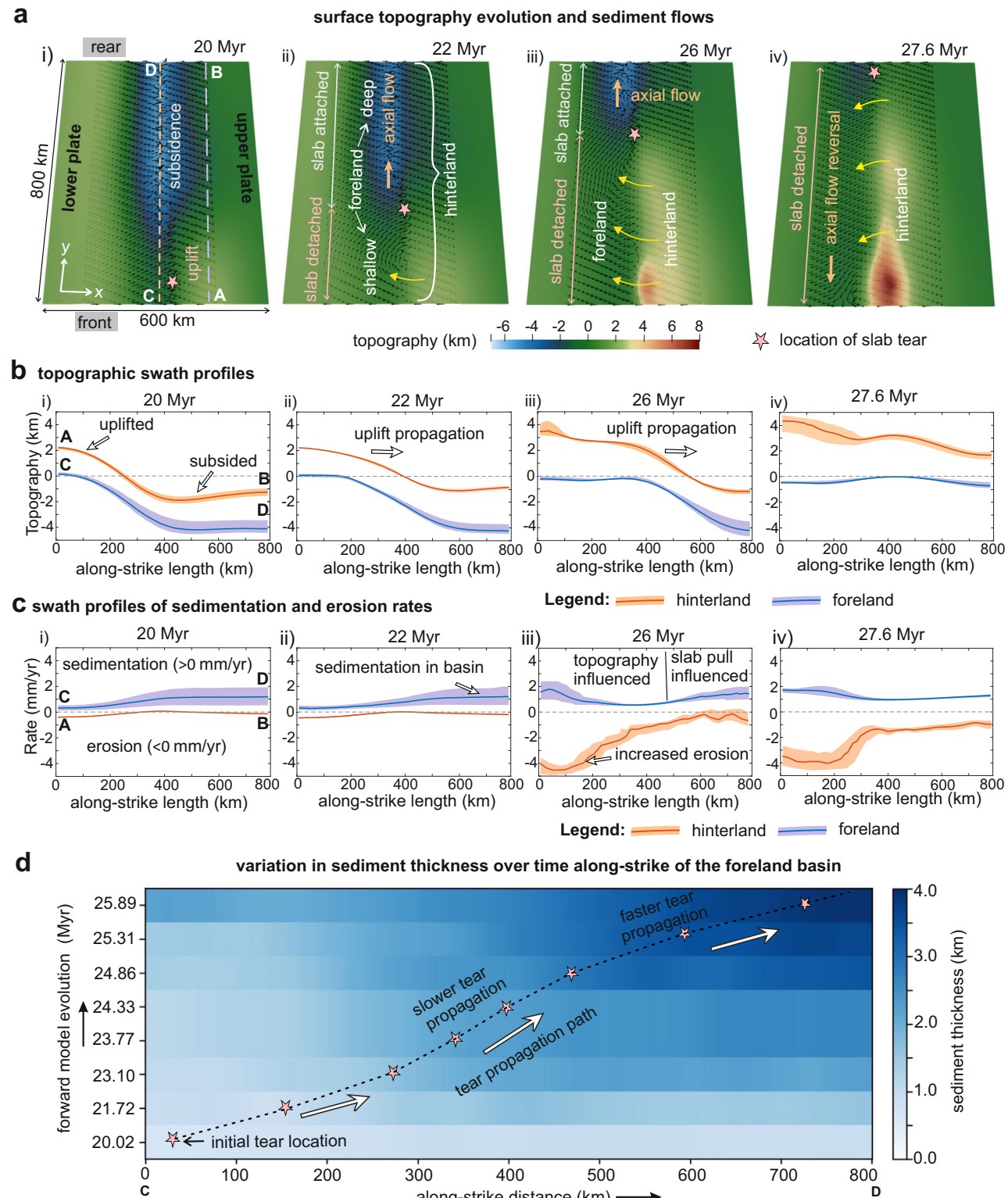

**Fig. 3 | Along-strike changes in surface topography, sediment transport, erosion, and sedimentation in the hinterland and foreland during slab-tear propagation. a** Evolution of surface topography and sediment transport direction on the model top surface (i.e., elevation = 0 km) as the slab tear propagates along-strike. **b** Swath topographic profiles taken along-strike of the hinterland (section AB) and foreland (section CD) are indicated in subpanels-i of (**a**). **c** Swath profiles of sedimentation and erosion rates taken along-strike of the hinterland (section AB) and foreland basin (section CD). **d** Variation of sediment thickness over time along-

strike of the foreland basin (section CD shown in subpanel-i of **a**). Note that sediment thickness increases along-strike of the foreland in the direction of slab-tear propagation, as there the slab remains attached for a longer duration, resulting in longer subsidence and larger accumulation of sediments till the tear reaches the rear end. Also, the tear propagation is not strictly linear. The overall along-strike tear-propagation velocity decreases during the vertical tearing phase, but accelerates during the terminal phase due to increased slab-pull forces from the hanging slab (Supplementary Fig. S3).

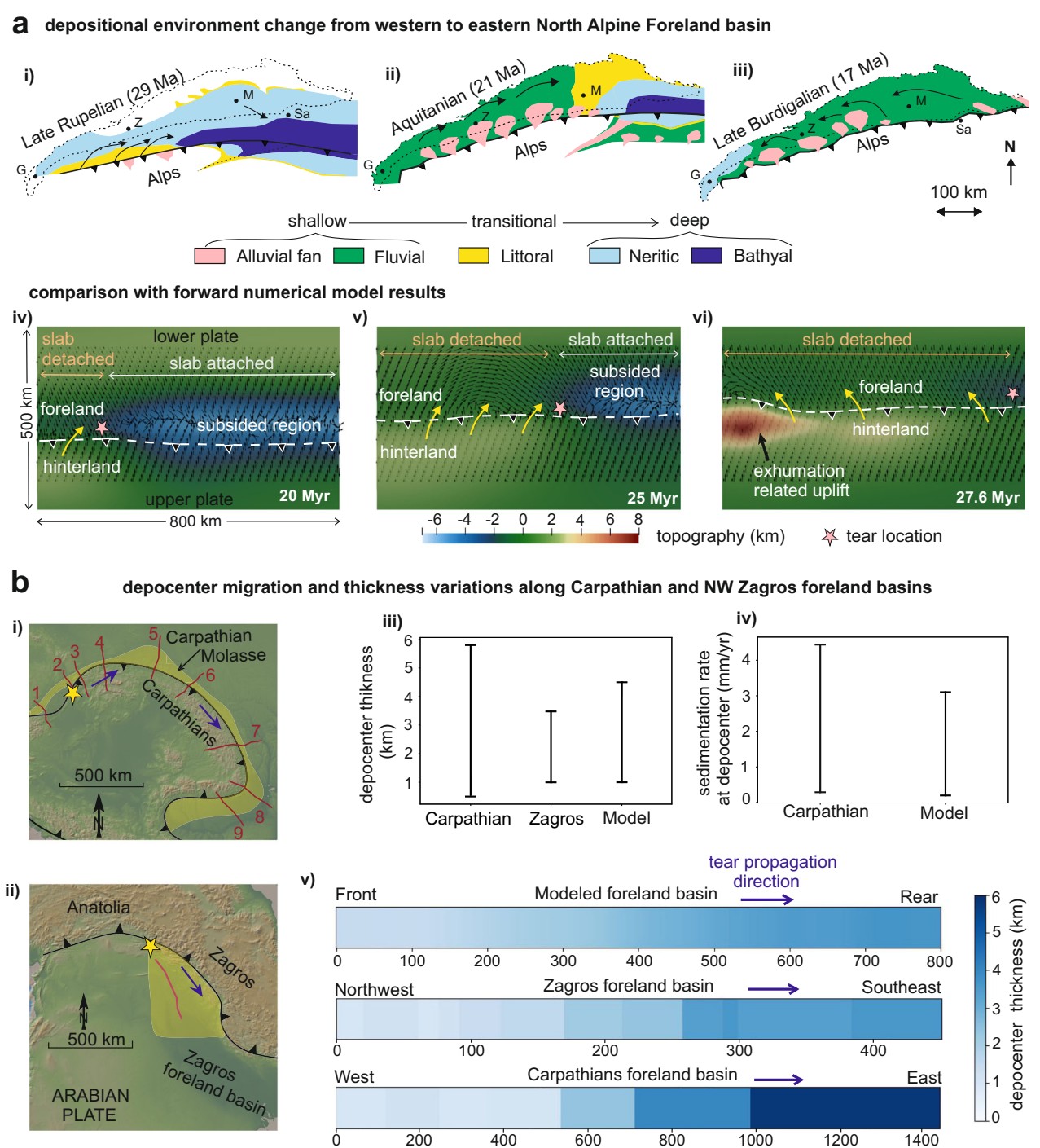

**Fig. 4 | Comparison of model-predicted slab-tearing signals with geological observations from foreland basins. a** Along-strike differential uplift/subsidence as observed in the NAFB during Late Oligocene (**i**) to Early Miocene time (**iii**). Note that the dominant sediment transport direction initially was from west-to-east. With time, this reversed to east-to-west drainage (modified from ref. 41. under CC-BY-NC license). The lateral propagation of slab tear in the numerical model (**iv–vi**) gives rise to similar along-strike differential uplift/subsidence, sediment flow path during slab-tear propagation, and its reversal upon tear completion. (**b i** and **ii**) show the transect locations (also see Supplementary Figs. S15 and S16) where

depocenter thickness and sedimentation rates are compiled in Carpathians[6] and Zagros foreland basin[11]. These data are compared with model results shown in (**iii–v**). (**iii** and **iv**) show comparison of foreland depocenter thickness (km) and sedimentation/accumulation rate (mm yr⁻¹) between Carpathian and model estimates. (**v**) shows a comparison of sediment thickness variation along-strike of the foreland basin. Note that both model and depocenter thickness of NW Zagros and Carpathian show that total accumulated sediment thickness in the foreland basin increases along the direction of slab-tear propagation (figure made with GeoMapApp[70] (www.geomapapp.org) / CC-BY).

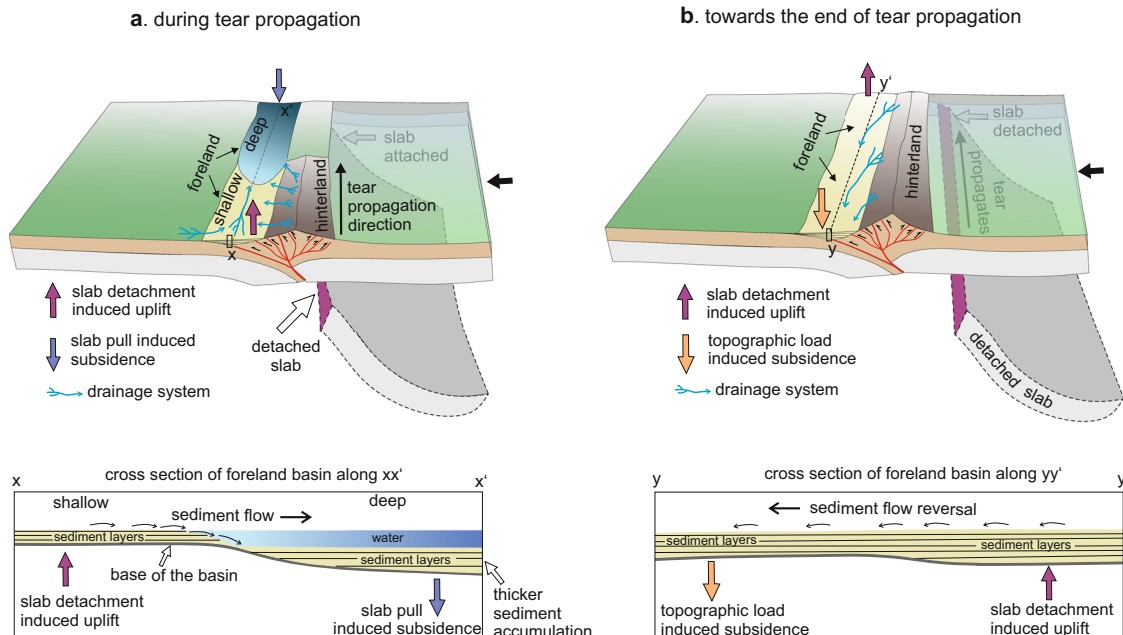

**Fig. 5 | Schematic summary of slab-tearing signals in a foreland basin. a** shows how tear propagation creates along-strike coeval but contrasting depositional environments, and the development of an axial-drainage system toward the still-attached slab region. These processes eventually create an along-strike difference in depositional thickness. **b** shows how the axial-drainage system within the basin reverses as soon as the slab-tear propagates to the other end, resulting in isostatic uplift in that region.

outpace erosion under the imposed constant-diffusivity parameter, high topography can still develop and persist despite active surface processes. Because erosional efficiency in natural landscapes varies through space and time with topographic steepness, drainage-network organization, climate, and subsurface rock properties[64]. So a constant diffusion coefficient in our model is unlikely to capture the whole system. Future models should therefore couple hillslope diffusion with a stream-power-based fluvial incision law, as commonly implemented in landscape-evolution models such as Landlab[65].

### Initial model setup and model parameters

We present ten numerical models (Supplementary Tables S2 and S3). The reference model (Model 1) domain is sized $1800 \times 800 \times 690$ km³ (Supplementary Fig. S1b), resolved by $453 \times 261 \times 181$ nodal points. Additionally, ~300 million randomly distributed markers are used for advecting material properties. The setup involves a moving continental overriding plate, a downgoing oceanic plate, and a stationary continental plate. The continental crust consists of 20 km of upper crust (wet quartzite rheology) and 18 km of lower crust (plagioclase rheology). The oceanic crust is represented by 2 km of basaltic upper crust and 5 km of gabbroic lower crust. Both the lithospheric and the sub-lithospheric mantle are composed of anhydrous peridotite (dry olivine rheology). A weak zone, defined by low plastic strength and wet olivine rheology, is placed along the active margin to initiate subduction. Free slip boundary conditions are defined on all model sides. To simulate the topographical evolution, a 20-km-thick 'sticky air' layer with a viscosity of $10^{18}$ Pa s and a density of 1 kg m⁻³ is placed on top of the model as the internal free surface. Free slip boundary conditions are set on the front and back boundaries. Thermal boundary conditions involve zero heat flux on all side boundaries, while constant temperature conditions are applied on the upper and lower boundaries.

The left-side continent remains stationary (representing the fixed Eurasia), while the right-side continent (representing the Adriatic plate) is pushed at 3 cm yr⁻¹ to initiate subduction of the Piemont–Ligurian oceanic domain. Closure of the Piemont–Ligurian

ocean led to the collision of the European passive margin with the Adriatic plate. Once continental collision begins, the convergence velocity is reduced to 0.75 cm yr⁻¹ to replicate the convergence velocity during Alpine collision stage (35−0 Ma)[5,66]. To incorporate along-strike variability in the structure of the European passive margin, a small microcontinent is placed adjacent to the European stationary continent at the frontal side of the model domain (Fig. 2a). The microcontinent represents the Briançonnais high (Supplementary Fig. S1a), which was originally separated from the European continental passive margin by the opening of the Valais ocean[38,39]. Paleogeographic reconstructions suggest an age difference of 40 Myr between Valais and Piemont−Ligurian ocean[38]. In the initial model setup, we assign an age of 10 Myr to the Valais Ocean and 50 Myr to the Piemont−Ligurian Ocean (Supplementary Fig. S1b). This configuration produces a 40 Myr along-strike age contrast in the adjacent oceanic lithosphere, consistent with paleogeographic reconstructions of the European passive margin. Because the exact along-strike extent of the Valais Ocean parallel to the European passive margin remains poorly constrained (Supplementary Fig. S1a), we assumed the age of the oceanic domain increases linearly from the frontal to the rear end of the passive margin (Supplementary Fig. S1b). Accordingly, the transition in oceanic age between the two domains is gradual rather than abrupt (Supplementary Figs. S2 and S24).

Furthermore, we ran several other models by varying initial oceanic plate ages along-strike of the passive margin (Supplementary Table S2), as well as plastic and ductile rheological parameters of the mantle, to evaluate their effects on model results (Supplementary Table S3). In particular, we increased the activation volume of the lithospheric mantle. The applied activation volume values are within the range of previously suggested and modeled values[62,63]. This increases ductile mantle viscosity and strengthens its pressure dependence, causing viscosity to increase more strongly with depth (Supplementary Fig. S25). Because slab breakoff is driven by the negative buoyancy of the sinking oceanic slab, an increase in mantle viscosity enhances resistance to slab sinking and therefore can affect slab breakoff and tear propagation. We also tested another way of

increasing mantle viscosity by modifying the plastic viscosity through changes in the initial and final internal friction coefficients (Supplementary Table S3). Decreasing the initial internal friction coefficient reduces plastic viscosity, whereas increasing the final internal friction coefficient reduces strain weakening and thereby increases the effective visco-plastic viscosity of the deformed, cold lithospheric mantle at high pressure and depth[21]. Furthermore, we conducted sensitivity tests on microcontinent size by increasing its length and width (Supplementary Figs. S17–S20). To evaluate the effect of the thermal structure of the microcontinent on slab detachment and tear propagation, we carried out two additional experiments in which the Moho temperature of the microcontinent was increased from 520 °C in the reference model to 700 °C and 850 °C (Supplementary Figs. S21 and S22). The reference-model Moho temperature is at the lower end of the typical continental Moho temperature range (450–850 °C[67,68]), so these additional models allow us to assess the sensitivity of the results to a warmer microcontinental crust.

## Data availability

All relevant data, model inputs, and outputs presented in this study are available via Zenodo at https://doi.org/10.5281/zenodo.20273784 (ref. 69) and in the Supplementary Information.

## Code availability

The numerical code I3ELVIS used for 3D numerical experiments, and all the visualization scripts to generate the figures are available via Zenodo at https://doi.org/10.5281/zenodo.20273784 (ref. 69). 3D data visualization was carried out by Paraview.

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

## Acknowledgements

The authors gratefully acknowledge the support of the state of Baden-Württemberg through bwUniCluster (2.0). All 3D numerical models

were run in bwUniCluster (2.0) HPC, and access was provided by the Karlsruhe Institute of Technology (KIT).

## Author contributions
G.M. conceptualized, designed, and conducted the numerical experiments and wrote the first draft of the manuscript. N.T. contributed to conceptualization, funding acquisition, data interpretation, and manuscript writing. A.B. assisted in designing the experiments and resolving numerical challenges. L.E. interpreted and compared model results with geological data. T.G. developed the original numerical code. All authors discussed the "Results" and "Methods" sections and contributed to data interpretation and manuscript preparation.

## Funding
N.T. discloses support for the research of this work from the Deutsche Forschungsgemeinschaft (DFG) [grant number TO 1364/1-1]. T.G. discloses support for the research of this work from the Swiss National Science Foundation (SNSF) [grant number 200021_192296] and the ILP Task Force "Bio-geodynamics of the Lithosphere". G.M., L.E., and A.B. declare no relevant funding. Open Access funding was enabled and organized by Projekt DEAL.

## Competing interests
The authors declare no competing interests.
