## [Transparent Peer Review file · Nature Communications]

Slab tearing and its surface signals controlled by passive margin strength

Corresponding Author: Dr Giridas Maiti

Version 0:

Reviewer comments:

Reviewer #1

(Remarks to the Author)

I read with interest the manuscript "Slab tearing and its surface signals controlled by passive margin strength". In this work the authors present a series of 3D numerical models of lateral propagation of slab detachment where variations in passive-margin strength control the initiation, propagation, and surface imprint of slab tearing. The numerical simulations presented in this work fit more realistically geological records of lateral slab detachment rates (of about 10 cm/yr). These estimates are an order of magnitude smaller compared with previous numerical simulations. Modeling results are consistent with tectonostratigraphic signatures of natural examples from the Alps, Carpathians and Zagros.

I have some comments that should be addressed before the final decision will be made:

- 1) Micro-continent: the authors introduce a small piece of continental plate (called micro-continent) at one side of the models. I would like to see how its properties (size, thermal structure) affect the slab detachment process.
- 2) In figure 3 the authors show a plot where one can see how tear propagates as a function of sediment thickness. This is a very useful plot. However, a good idea is to have several plots like these (based on the model runs presented, or if necessary, on new model runs) where one can see different scenarios where different sedimentary thickness affect slab detachment.
- 3) I made some additional suggestions on the annotated manuscript.

Reviewer #2

(Remarks to the Author)

I have read with great interest the submitted manuscript entitled "Slab tearing and its surface signals controlled by passive margin strength" by Maiti et al. In their work the authors carry out a set of three-dimensional forward geodynamical simulations of subduction and slab tearing while also investigating the effect of such processes at the surface.

The type of modelling is state-of-the-art and the coupling of 3D modelling with surface processes simulations which has become more and more prevalent in the last 2 decades has enabled the computational geodynamics community to tackle feedback mechanisms between deep(er) and shallow(er) processes. The numerical codes used in this work have been used in many studies and benchmarked against other codes, so that I have no comment with regards to this choice.

I found the article particularly well written and structured. Figures are clear, and the supplementary material is systematic and helpful. As such I do not have many comments. I will however mention the peculiar viscosity color scale in virtually all figures, that goes from 10^{22} to 10^{23} in one 'tick' and then to 10^{24} in 9 ticks.

Also, looking at Fig.3a for example, it looks like a substantial area of the topography is at +8km (or more?) despite the presence of surface processes eroding this topography.

Another minor comment is the repetition in the Methods section of 'mass momentum and energy equations' within 10 lines of each other. Also, I would use a dot in eq 10 to indicate that grad h is dotted with the velocity vector.

At line 227, I find that the sentence does not read too well (I grouped 'predict once' together when reading it), may be formulate as 'Models predict that once slab-tear propagation is completed then isostatic ...'?

I will finish with a comment on the bibliography that is rather 'messy': some references have a doi, some do not. Some author lists are 'et al', others are listed in full. Some volume numbers are bold, some are not. Some have page numbers, some don't. Some journals are abbreviated others are not. Refs 1 and 4 are full of typos. An obvious easy fix.

Finally, I will say that the modelling is elegant and satisfactorily address the question at hand with a well structured Discussion section.

Reviewer #3

(Remarks to the Author)

In this paper, Maiti and co-authors aim at understanding the parameters controlling slab tear migration rates and to address the apparent discrepancy between rates inferred from sedimentological observations and those predicted by numerical models, which are generally much higher than observed.

The authors use a three-dimensional numerical model consisting of a subducting oceanic plate attached to a continental domain, incorporating a microcontinent at the front side of the domain, and/or a rheological contrast within the oceanic plate along the passive margin. The presence of a microcontinent induces earlier slab detachment relative to the rest of the domain, followed by horizontal tear propagation. The parametric study investigates the influence of the thermal age of the oceanic lithosphere, viscous rheology, and friction coefficient within the subducting mantle. Surface processes, including erosion and sedimentation, are also taken into account. The simulations are performed using the thermo-mechanical code I3ELVIS, which solves the mass, momentum, and energy conservation equations for incompressible media using a finite-difference scheme with non-linear visco-plastic rheologies. The model is coupled to a diffusion equation with a source term representing uplift rate, as well as horizontal advection of topography, to account for tectonic and isostatic processes. The results indicate that increasing the contrast in thermal age of the oceanic lithosphere, the activation volume of viscous creep, or the contrast in friction coefficients within the oceanic mantle strongly affects slab tear propagation rates. These effects are sufficient to reduce propagation rates to commonly accepted values inferred from sediment accumulation rates observed in natural foreland basins. In addition, the study highlights how subsidence in sedimentary basins changes from slab-driven (when the slab is attached) to topography-driven (when topography grows after slab detachment), and how range-parallel drainage can develop as a result of the topographic contrast between slab-detached and slab-attached lower plates. The authors further show that range-parallel drainage reversal may occur toward the end of the slab tearing process, driven by uplift in the area where the slab was last attached. This study also helps explain why different depositional environments (e.g., shallow marine or continental, versus deep marine) can coexist in closely spaced foreland basins. The authors analyze their results in the context of natural examples from the Alps–Carpathians–Zagros region, where depocenter migration associated with slab tear propagation has been inferred.

Overall, I believe this paper presents interesting and potentially significant results that should deserve publication in a journal such as Nature Communications. Their interpretation on the causes of subsidence, depocenter migration and sedimentological facies variations in space and time in foreland basins is particularly interesting. However, several aspects would benefit from clearer presentation and more thorough discussion. In my view, the main limitation is not the scientific relevance of the conclusions but rather the transparency of the model setup and assumptions.

- First, it is difficult to clearly understand how rheological contrasts within the oceanic domain are implemented. In models that include a microcontinent, I assume that the contrast corresponds to the difference between the narrow oceanic domain and the larger one, as illustrated in the example of the Valaisan domain in Figure S1A. However, in the two models without a microcontinent, it is unclear how this contrast is prescribed. Is it implemented as a linear gradient between the frontal and rear parts of the domain, or as an abrupt lateral change? In the last case, where is it located exactly? When I read “passive-margin strength variations controlling both the rate of propagation and the migration of surface responses...” I expect a model where the passive margin has several domains or margin-perpendicular stripes of different rheologies, and I would like to see the propagation rate in each domain. It’s not what is presented here since each model involves only 2 different rheologies. I have no problem with this approach, but we need more details on the initial conditions.
- Regarding the chosen implementation of rheological contrasts, is the thermal age difference between oceanic domains still preserved at the onset of slab tearing (i.e., ~20 Ma after the beginning of the model)? I would expect such contrasts to dissipate relatively rapidly. In addition, the rationale for modifying the activation volume of viscous creep should be clarified, given that this parameter enters the pressure-dependent term of the exponential flow law, and slab tear occurs at relatively shallow upper mantle levels here.
- Based on Figure 3d, slab tear propagation appears to be linear in time in the reference model. What controls this behavior? If two oceanic lithospheres with contrasting rheologies are present, one might expect a two-phase propagation (e.g., an initial fast phase followed by a slower one, reflecting front side to rear side slab tear propagation). Or maybe is the initial detachment phase in the smaller oceanic domain simply too short-lived to be seen? This again highlights the difficulty in identifying precisely where the rheological contrast is applied.
- In addition, when examining the forces exerted by slab pull on the subducting plate, the authors show that as slab detachment progresses, the force acting on the still-attached portion of the slab increases exponentially, which is physically intuitive (Figure S3). However, this raises the question of why slab tear propagation remains linear in time. Is there no direct relationship between slab pull and slab tearing, or does a negative feedback mechanism operate to prevent acceleration of the tear propagation? This point would benefit from further discussion.

Note that another paper by the same first authors (and 2 similar co-authors) was published in JGR in 2024* and presents a parametric study of slab tear propagation as a function of collision obliquity. The present study explores a different tectonic setting, focusing on frontal collision, with non-cylindricity introduced only by a small continental block located ahead of the main continent (and/or by a rheological contrast) on the front side of the model domain.

When these results are considered together with those of the 2024 study, it appears that slab tear propagation is controlled by a wide range of factors, including the presence of a microcontinent, collision obliquity, and rheological contrasts along the passive margin. In this context, it is not entirely clear why the present study emphasizes this specific factor as the dominant control. It might be valuable to broaden the discussion and propose a more general synthesis of the primary parameters governing slab tear propagation rates. Given the complementary nature of the two studies, the authors are in a strong

position to draw more general and potentially more impactful conclusions.

*Maiti, G., Koptev, A., Baviile, P., Gerya, T., Crosetto, S., & Andrić-Tomašević, N. (2024). Topography response to horizontal slab tearing and oblique continental collision: Insights from 3D thermomechanical modeling. *Journal of Geophysical Research: Solid Earth*, 129, e2024JB029385.

Version 1:

Reviewer comments:

Reviewer #1

(Remarks to the Author)

The authors answered satisfactory my previous concerns and I am satisfied with the revision process.

Reviewer #3

(Remarks to the Author)

I am satisfied with the responses and the important round of modifications made by the authors, and I have no further comments or suggestions. Thank you !

Reviewer #1 (Remarks to the Author):

I read with interest the manuscript "Slab tearing and its surface signals controlled by passive margin strength". In this work the authors present a series of 3D numerical models of lateral propagation of slab detachment where variations in passive-margin strength control the initiation, propagation, and surface imprint of slab tearing. The numerical simulations presented in this work fit more realistically geological records of lateral slab detachment rates (of about 10 cm/yr). These estimates are an order of magnitude smaller compared with previous numerical simulations. Modeling results are consistent with tectonostratigraphic signatures of natural examples from the Alps, Carpathians and Zagros.

I have some comments that should be addressed before the final decision will be made:

1) Micro-continent: the authors introduce a small piece of continental plate (called micro-continent) at one side of the models. I would like to see how its properties (size, thermal structure) affect the slab detachment process.

Response: We thank you for this insightful suggestion and have now systematically tested the effects of microcontinent length, width, and thermal structure on slab detachment and tear propagation.

To evaluate the influence of microcontinent length on slab tearing and its surface expression, we ran two additional models in which the along-strike length of the microcontinent (y-direction) was varied relative to the reference model. In the reference model, the microcontinent is 320 km long, corresponding to 40% of the model dimension in the y-direction (i.e., 800 km). We therefore tested one model with a longer microcontinent of 480 km (60%) and one with a shorter microcontinent of 160 km (20%). The results show that a longer microcontinent produces tearing behaviour broadly similar to the reference model, whereas a shorter microcontinent delays slab breakoff, tear initiation, and along-strike tear propagation. In the shorter case, collision-related topography develops before tearing fully propagates, making the surface expression of slab tearing less distinct. These results are now shown in Supplementary Figs. S17 and S18 and discussed in the revised main text (Lines 168-171).

Similarly, to assess the effect of microcontinent width (x-direction), we carried out two additional models in which the width of the microcontinent was increased relative to the reference model. In the reference model, the microcontinent is 40 km wide. We therefore tested widths of 60 km and 80 km. In both cases, slab breakoff at the frontal end initiates at approximately the same time as in the reference model (~20 Myr). However, increasing microcontinent width delays the along-strike propagation of the slab tear. This is because a wider microcontinent prolongs the phase of vertical tearing at its rear edge, where the tear transitions from the microcontinent to the adjacent oceanic domain. The delayed tear propagation is also reflected at the surface response, where the rear foreland basin remains subsided for a longer duration. These results show that microcontinent width has little effect on the timing of slab breakoff initiation, but it does influence the duration and geometry of tear propagation, as well as the persistence of the associated foreland basin signal. We have added these results as Supplementary Figs. S19 and S20 and discussed in the revised main text (Lines 171-173; 277-279).

To investigate the effect of microcontinent's thermal structure on slab detachment and tear propagation, we carried out two additional experiments in which the Moho temperature of the microcontinent was increased from 520 °C in the reference model to 700 °C and 850 °C. The reference-

model value lies at the lower end of the typical continental Moho temperature range (450–850 °C), and these additional models therefore allow us to assess the sensitivity of the results to a warmer micro-continental crust (Lines 444-450). The simulations show that increasing Moho temperature promotes stronger intraplate deformation within the microcontinent and enhances vertical thickening as it approaches the subduction zone, likely in response to collision-induced stress transmission. Slab necking begins at approximately the same time as in the reference model (~20 Myr), but final breakoff at the frontal end takes longer to develop, which in turn delays the along-strike propagation of the slab tear towards the rear end of the model. In the reference model, the tear reaches the rear end at ~28 Myr, whereas in the warmer models it arrives later, at ~32 Myr. These results and their interpretation have now been added to the Supplementary Information as Supplementary Figs. S21 and S22 and discussed in the revised main text (Lines 173-176).

2) In figure 3 the authors show a plot where one can see how tear propagates as a function of sediment thickness. This is a very useful plot. However, a good idea is to have several plots like these (based on the model runs presented, or if necessary, on new model runs) where one can see different scenarios where different sedimentary thickness affect slab detachment.

Response: We thank you for this valuable suggestion. We have now generated the same type of plots for end-member cases for models with the fastest tear propagation (Model_3) and the slowest tear propagation (Model_4), and compared them with the reference model (Supplementary Fig. S23). It is to be noted that in our models sediment-thickness variations are primarily a consequence of slab-tear propagation duration and it is not an imposed factor controlling slab detachment. Therefore, in these additional plots we illustrate how different tear-propagation scenarios influence the magnitude and duration of along-strike sediment-thickness variations.

The comparisons show that this effect depends strongly on tear-propagation velocity and duration. In the fast-propagation case (e.g., 26.6 cm yr⁻¹), along-strike differences in sediment thickness remain relatively small and exist only for a short time, because the tear migrates rapidly along-strike, and thereafter collisional overprints start occurring (Supplementary Fig. S23b). In contrast, in the slow-propagation case (e.g., 5.97 cm yr⁻¹), along-strike sediment-thickness differences become much more pronounced and remain for a substantially longer time (Supplementary Fig. S23c). The reference model, with an intermediate tear-propagation rate (10.5 cm yr⁻¹), shows intermediate behaviour between these two end members. We have now included these additional plots in Supplementary Fig. S23 to demonstrate how sediment-thickness variations evolve under contrasting slab-tearing scenarios and discussed it in the main text (Lines 179-182; 252-256). Models with intermediate tear-propagation rates will show the same general relationship, where the magnitude and duration of along-strike sediment-thickness differences will depend on tear-propagation duration and velocity.

3) I made some additional suggestions on the annotated manuscript.

Response: We thank you for the additional suggestions provided in the annotated manuscript. We have addressed each point as follows.

Comment on Fig 2a: What do the "x" and "y" symbols represent in this rear cut?

Response: To show the evolution of the slab at the rear side, we provide a cross section taken at the rear edge of the model domain. The “x” and “y” symbols were originally used to indicate the location of this cross section. For clarity, we have now replaced them with the label section XX’.

Comment on Fig. 2a: What are the constraints about the size of the micro-continent? What are the effects of micro-continent dimensions (in x and y) on the lateral propagation of slab tearing?

Response: The size of the micro-continent is constrained based on palaeotectonic reconstructions of the Adria–Eurasia collision. We have provided the corresponding palaeotectonic reconstruction as Supplementary Fig. S2, together with the relevant references in the main text (Lines 416-420). The effects of micro-continent dimensions in the x and y directions on the lateral propagation of slab tearing are also discussed in our response to Comment 1.

Comment on Fig 2j: Specify in figure caption the values of the white isocurves.

Response: We have now included the values of the white isocurves directly in the figure.

Comment on Fig. 3a: Reduce the number of arrows here (a - panels) for a better visualization.

Response: We tested some visualizations with reduced number of arrows. However, this made the flow paths less clear and did not serve appropriately the purpose of this figure. We therefore retained the original arrow density.

Comment on Fig. 3d: I suggest preparing a couple of figures like this (possibly more model runs are required) where sediment thickness accumulated is different, i.e., higher or smaller than the one presented here. So, the reader can better understand the effect of sedimentation on lateral tear propagation.

Response: We have added new figures in the revised manuscript to address this point (Supplementary Fig. S23). The effect of sediment thickness on lateral tear propagation is discussed in detail in our response to Comment 2.

Reviewer #2 (Remarks to the Author):

I have read with great interest the submitted manuscript entitled "Slab tearing and its surface signals controlled by passive margin strength" by Maiti et al. In their work the authors carry out a set of three-dimensional forward geodynamical simulations of subduction and slab tearing while also investigating the effect of such processes at the surface.

The type of modelling is state-of-the-art and the coupling of 3D modelling with surface processes simulations which has become more and more prevalent in the last 2 decades has enabled the computational geodynamics community to tackle feedback mechanisms between deep(er) and shallow(er) processes. The numerical codes used in this work have been used in many studies and benchmarked against other codes, so that I have no comment with regards to this choice.

I found the article particularly well written and structured. Figures are clear, and the supplementary material is systematic and helpful. As such I do not have many comments. I will however mention the peculiar viscosity color scale in virtually all figures, that goes from 10^{22} to 10^{23} in one 'tick' and then to 10^{24} in 9 ticks.

Response: We thank you for the positive and encouraging assessment of our manuscript, and for highlighting the clarity of the presentation, figures, and supplementary material. We very much appreciate the reviewer's observation regarding the viscosity colour scale. We have now changed the viscosity colour scale that goes from 10^{18} to 10^{24} . All the related figures have been updated accordingly.

Also, looking at Fig.3a for example, it looks like a substantial area of the topography is at +8km (or more?) despite the presence of surface processes eroding this topography.

Response: We thank you for raising this important point. In our models, surface processes are simulated using a linear diffusion equation with a constant diffusion coefficient throughout the simulation. The locally high elevation in Fig. 3a show that, for a short period, tectonically driven rapid surface uplift outpaces diffusive smoothing. However, the topography gets smoothed gradually.

We agree that this is a simplification. In nature, erosional efficiency varies in space and time as a function of slope, drainage development including water discharge, climate, and subsurface rock properties, so a constant diffusion coefficient likely does not capture the whole system. Following your comment, we ran additional models with a higher diffusion coefficient and it produces lower surface topography, confirming the sensitivity of relief to this parameter. Most importantly, none of our inferences and conclusions changed during this sensitivity analysis. In the future a more complex treatment would involve spatially and/or temporally variable erosional efficiency, but this is beyond the scope of our upper mantle-scale study. We have now revised the Methods section to clarify this point and explicitly acknowledge this limitation in the discussion of our modelling approach (Lines 382-391). A more realistic treatment in future work should include coupling hillslope diffusion with a stream-power-based fluvial incision law, as commonly implemented in landscape-evolution models.

Another minor comment is the repetition in the Methods section of 'mass momentum and energy equations' within 10 lines of each other. Also, I would use a dot in eq 10 to indicate that $\text{grad } h$ is dotted with the velocity vector.

Response: We thank you for this careful observation. We have now removed the repetition in the Method section and used a dot in Eq.10 to indicate that $\text{grad } h$ is dotted with the velocity vector.

At line 227, I find that the sentence does not read too well (I grouped 'predict once' together when reading it), may be formulate as 'Models predict that once slab-tear propagation is completed then isostatic ... '?

Response: We have now modified the text. Thanks.

I will finish with a comment on the bibliography that is rather 'messy': some references have a doi, some do not. Some author lists are 'et al', others are listed in full. Some volume numbers are bold, some are not. Some have page numbers, some don't. Some journals are abbreviated others are not. Refs 1 and 4 are full of typos. An obvious easy fix.

Response: We have carefully revised the bibliography. In accordance with Nature Communications style, all authors are listed unless a reference has six or more authors, in which case the first author is followed by "et al.". Therefore some author lists are still 'et al'. We have fixed all other problems and typos. We thank the reviewer for highlighting these issues.

Finally, I will say that the modelling is elegant and satisfactorily address the question at hand with a well-structured Discussion section.

Response: We thank you for this positive evaluation of our modelling approach and discussion.

Reviewer #3 (Remarks to the Author):

In this paper, Maiti and co-authors aim at understanding the parameters controlling slab tear migration rates and to address the apparent discrepancy between rates inferred from sedimentological observations and those predicted by numerical models, which are generally much higher than observed.

The authors use a three-dimensional numerical model consisting of a subducting oceanic plate attached to a continental domain, incorporating a microcontinent at the front side of the domain, and/or a rheological contrast within the oceanic plate along the passive margin. The presence of a microcontinent induces earlier slab detachment relative to the rest of the domain, followed by horizontal tear propagation. The parametric study investigates the influence of the thermal age of the oceanic lithosphere, viscous rheology, and friction coefficient within the subducting mantle. Surface processes, including erosion and sedimentation, are also taken into account. The simulations are performed using the thermo-mechanical code I3ELVIS, which solves the mass, momentum, and energy conservation equations for incompressible media using a finite-difference scheme with non-linear visco-plastic rheologies. The model is coupled to a diffusion equation with a source term representing uplift rate, as well as horizontal advection of topography, to account for tectonic and isostatic processes.

The results indicate that increasing the contrast in thermal age of the oceanic lithosphere, the activation volume of viscous creep, or the contrast in friction coefficients within the oceanic mantle strongly affects slab tear propagation rates. These effects are sufficient to reduce propagation rates to commonly accepted values inferred from sediment accumulation rates observed in natural foreland basins. In addition, the study highlights how subsidence in sedimentary basins changes from slab-driven (when the slab is attached) to topography-driven (when topography grows after slab detachment), and how range-parallel drainage can develop as a result of the topographic contrast between slab-detached and slab-attached lower plates. The authors further show that range-parallel drainage reversal may occur toward the end of the slab tearing process, driven by uplift in the area where the slab was last attached. This study also helps explain why different depositional environments (e.g., shallow marine or continental, versus deep marine) can coexist in closely spaced foreland basins. The authors analyze their results in the context of natural examples from the Alps–Carpathians–Zagros region, where depocenter migration associated with slab tear propagation has been inferred.

Overall, I believe this paper presents interesting and potentially significant results that should deserve publication in a journal such as Nature Communications. Their interpretation on the causes of subsidence, depocenter migration and sedimentological facies variations in space and time in foreland basins is particularly interesting. However, several aspects would benefit from clearer presentation and more thorough discussion. In my view, the main limitation is not the scientific relevance of the conclusions but rather the transparency of the model setup and assumptions.

Response: We thank you for this thoughtful and constructive assessment of our work. We also appreciate the reviewer's important observation that the presentation of the model setup and assumptions could be made clearer. In response, we have revised the manuscript to improve the model setup section and modelling strategy (Lines 409-428). We also clarify the key assumptions behind parameter choices, and expand the discussion where needed (Lines 429-450).

- First, it is difficult to clearly understand how rheological contrasts within the oceanic domain are implemented. In models that include a microcontinent, I assume that the contrast corresponds to the difference between the narrow oceanic domain and the larger one, as illustrated in the example of the Valaisan domain in Figure S1A. However, in the two models without a microcontinent, it is unclear how this contrast is prescribed. Is it implemented as a linear gradient between the frontal and rear parts of the domain, or as an abrupt lateral change? In the last case, where is it located exactly? When I read “passive-margin strength variations controlling both the rate of propagation and the migration of surface responses...” I expect a model where the passive margin has several domains or margin-perpendicular stripes of different rheologies, and I would like to see the propagation rate in each domain. It’s not what is presented here since each model involves only 2 different rheologies. I have no problem with this approach, but we need more details on the initial conditions.

Response: We thank you for raising this important point. In our models, passive-margin strength variations are primarily introduced through along-strike differences in the age of the adjacent oceanic lithosphere. In the reference model and in the other models that include a microcontinent, we impose a linear increase in oceanic age from the frontal to the rear end (Supplementary Fig. S1b). We adopted this formulation because the exact along-strike extent of the Valais Ocean parallel to the European passive margin is not well constrained by current paleogeographic reconstructions (Supplementary Fig. S1a). The linear gradient therefore provides a simplified but geologically motivated way to represent along-strike strength variations in the oceanic domain. Thus, the rheological contrast is not implemented as an abrupt lateral step, but as a continuous along-strike gradient. In these models, the microcontinent introduces an additional heterogeneity superimposed on this background strength gradient, which further influences the localization of slab breakoff and the subsequent tear propagation. We have revised the Model setup section to make this distinction clearer in the method section (Lines 416-428).

In the models without a microcontinent, the age of the oceanic domain also increases linearly from the frontal to the rear end of the model, such that lithospheric strength also increases progressively in the same direction. Numerically, this is prescribed by assigning oceanic ages at the four corners of the oceanic domain (i.e., at the two frontal corners and the two rear corners; Supplementary Fig. S1b), from which the age field varies linearly across the domain (Supplementary Figs. S2 and S24). This configuration promotes initial slab breakoff in the frontal, weaker part of the model and subsequent along-strike propagation towards the stronger rear part. We have now clarified this more explicitly in the description of the initial conditions and revised the corresponding text and figures accordingly.

- Regarding the chosen implementation of rheological contrasts, is the thermal age difference between oceanic domains still preserved at the onset of slab tearing (i.e., ~20 Ma after the beginning of the model)? I would expect such contrasts to dissipate relatively rapidly. In addition, the rationale for modifying the activation volume of viscous creep should be clarified, given that this parameter enters the pressure-dependent term of the exponential flow law, and slab tear occurs at relatively shallow upper mantle levels here.

Response: We thank you for this important point. In our model, the thermal age difference between oceanic domains results in an along-strike variation in slab thickness and slab strength. We measured

slab thickness at the start of the model run and before the onset of slab tearing. Although the slab thickens during model evolution, and the thickness ratio of courses slightly decreases, the relative along-strike thickness contrast is still notably preserved up to the onset of tearing (Supplementary Fig. S24). We have now added a slab thickness plot to the revised manuscript showing that this contrast remains present before slab tearing initiates (Supplementary Fig. S24).

In our models, detachment (slab break-off) of the oceanic slab from the continental lithosphere is driven by the weight of the sinking oceanic slab into the mantle. Therefore, we agree that mantle viscosity exerts an important control in slab-sinking. Increasing the activation volume of the mantle increases its ductile viscosity. The viscosity increases with pressure and therefore with depth. This leads to greater viscous resistance to slab sinking. Therefore, although slab tear occurs at relatively shallow upper-mantle levels, the deeper part of the oceanic slab is more strongly supported by the surrounding high-viscosity mantle, which slows slab sinking and hence reduces tear propagation velocity. The applied activation volume values are within the range of previously suggested and modelled values (cf. Ranalli 1995; Turcotte and Schubert 2014). We have now added these explanations to the Method section for clarity (Lines 432-443) and added a new supplementary Fig. S25 to show how activation volume of viscous creep affects the mantle viscosity.

- Based on Figure 3d, slab tear propagation appears to be linear in time in the reference model. What controls this behavior? If two oceanic lithospheres with contrasting rheologies are present, one might expect a two-phase propagation (e.g., an initial fast phase followed by a slower one, reflecting front side to rear side slab tear propagation). Or maybe is the initial detachment phase in the smaller oceanic domain simply too short-lived to be seen? This again highlights the difficulty in identifying precisely where the rheological contrast is applied.

Response: We thank you for this important observation. We have conducted a detailed analysis of slab tear propagation. We find that the propagation is not strictly linear with time. In the reference model, as well as other models containing a micro-continent, slab tear initially localizes in the micro-continent region and propagates relatively fast. Thereafter, during the transition of tearing from micro-continent to the oceanic domain the tear propagation slows down relatively and it is accompanied by a short phase of vertical tearing. In the last phase, when the tear propagates only through the oceanic domain it accelerates, primarily because slab pull increases as tearing progresses. We have revised Figure 3d to better illustrate this non-linear behaviour.

In models without a micro-continent, the rheological or strength contrast exists only within the oceanic domain and increases linearly from the frontal to the rear end (Supplementary Fig. S24). In these cases, tear propagation also tends to accelerate during the final stage, again due to increasing slab pull. To make this clearer, we have prepared additional supplementary figure (Supplementary Fig. S23), similar to Figure 3d, for a model without micro-continent.

- In addition, when examining the forces exerted by slab pull on the subducting plate, the authors show that as slab detachment progresses, the force acting on the still-attached portion of the slab increases exponentially, which is physically intuitive (Figure S3). However, this raises the question of why slab tear propagation remains linear in time. Is there no direct relationship between slab pull and slab tearing, or does a negative feedback mechanism operate to prevent acceleration of the tear propagation? This point would benefit from further discussion.

Response: We thank you for raising this point. Indeed, in the last phase, when the tear propagates only through the oceanic domain it accelerates, primarily because slab pull increases as tearing progresses. We have revised Figure 3d to better illustrate this non-linear behaviour and revised the corresponding figure caption (Lines 141-144). Also a new figure in the supplementary figure (Supplementary Fig. S23) is added, that reflects the behaviour of slab-pull induced acceleration in tearing at the terminal phase.

- Note that another paper by the same first authors (and 2 similar co-authors) was published in JGR in 2024* and presents a parametric study of slab tear propagation as a function of collision obliquity. The present study explores a different tectonic setting, focusing on frontal collision, with non-cylindricity introduced only by a small continental block located ahead of the main continent (and/or by a rheological contrast) on the front side of the model domain.

When these results are considered together with those of the 2024 study, it appears that slab tear propagation is controlled by a wide range of factors, including the presence of a microcontinent, collision obliquity, and rheological contrasts along the passive margin. In this context, it is not entirely clear why the present study emphasizes this specific factor as the dominant control. It might be valuable to broaden the discussion and propose a more general synthesis of the primary parameters governing slab tear propagation rates. Given the complementary nature of the two studies, the authors are in a strong position to draw more general and potentially more impactful conclusions.

*Maiti, G., Koptev, A., Baille, P., Gerya, T., Crosetto, S., & Andrić-Tomašević, N. (2024). Topography response to horizontal slab tearing and oblique continental collision: Insights from 3D thermomechanical modeling. *Journal of Geophysical Research: Solid Earth*, 129, e2024JB029385.

Response: We thank you for this very important and constructive suggestion. We agree that slab-tear propagation is controlled by multiple factors, including rheological contrasts along the passive margin, the presence of a microcontinent, and collisional obliquity. In our earlier study (Maiti et al., 2024), we showed that presence of microcontinent and margin obliquity can both reduce tear-propagation rates. However, the predicted tear velocities still remained higher than those inferred from geological proxies, possibly because those models assumed laterally homogeneous passive margins. Moreover, margin obliquity may vary along strike and may be absent entirely in some collisional systems. In the present study, we focus specifically on non-oblique collisional margins and investigate how passive-margin strength variations arising from along-strike differences in oceanic age and the presence of a microcontinent influence tear propagation and its surface expression. These factors introduce inherited rheological heterogeneity into the subducting plate and adjacent passive margin, which strongly modulates the lateral propagation of slab tearing. By incorporating these along-strike strength variations together with a microcontinent, our models yield tear-propagation velocities that are closer to geological estimates from the Alps. This suggests that realistic tear propagation requires accounting for inherited passive-margin heterogeneity, rather than treating the margin as laterally uniform. Therefore, the aim of the present study is not to isolate a single universal control on slab tearing, but to demonstrate that passive-margin strength heterogeneity, particularly that arising from along-strike oceanic-age variations and microcontinental structure, is an important factor that should be considered alongside previously recognised controls such as collisional obliquity.

Following your suggestion, we have revised the Discussion to provide a broader synthesis of the main parameters governing slab-tear propagation (Lines 288-301). We now explicitly frame passive-margin strength heterogeneity, microcontinental architecture, and collisional obliquity as complementary controls whose relative importance may vary between collisional settings. We believe this revision strengthens the broader implications of the study and better links the present results to those of Maiti et al. (2024).